# Determinants of Smallholder Rice Farmers' Willingness-to-Pay for Private Extension Services in Liberia: The Case of Gibi District

Togba V. Sumo *, Cecilia Ritho  and Patrick Irungu 

Department of Agricultural Economics, University of Nairobi, Nairobi P.O. Box 29053-00625, Kenya;
ceciliaritho@gmail.com (C.R.); patrickirungu@yahoo.com (P.I.)
* Correspondence: togbavsumo@gmail.com

**Abstract:** Globally, many policymakers and extension professionals have advocated for the privatization of extension services in order to reduce the burden of funding faced by the state as well as to adequately respond to the low productivity problem of farmers as they endeavor to tackle productivity problems. This study assessed willingness-to-pay (WTP) for private extension services by farmers and identified the determinants of their WTP using Gibi District of Liberia as a case study. A multistage sampling technique was used in selecting 296 smallholder rice farmers in the district while the double-bounded dichotomous choice contingent valuation method was used to elicit maximum WTP value for farmers. Descriptive statistics were computed and the double-bounded logit model used to analyze the data. The findings revealed that 78.7% of the rice farmers were willing to pay for privatized extension services and on average, a farmer was willing to pay US$11.21 per farm visit, almost twice the average daily wage rate of a skilled worker in Liberia. The results from the model showed that WTP was significantly positively influenced by the household head's age, years of schooling, household size, annual income, and distance to extension service provider. The study recommends that the Liberian government and its development partners should encourage the private sector to invest more in extension services to take advantage of the relatively high farmers' WTP and effective demand. In addition, the government should design and implement programs that reduce transaction costs in addition to increasing farmers' income in order to enhance their capacity to pay for privatized extension services.

**Keywords:** contingent valuation; double-bounded model; private extension services; Liberia; willingness-to-pay

## 1. Introduction

Agricultural development relies largely on extension services to disseminate relevant information and skills on new technologies to farmers so that they can make their adoption decision [1]. Accordingly, agricultural extension is one of the major inputs with the potential of improving agricultural productivity, increase farmers' income, and improve their welfare [2,3]. Al-Zahrani et al. [4] note that agricultural extension is the most effective and critical means for disseminating agricultural information in order to influence farmers' behavior in making better decisions in adopting new technologies. Towards the end of the twentieth century, some Sub-Saharan African (SSA) countries renewed their commitment to reform their agricultural extension systems to provide farmer-centered, participatory, and demand-driven services [5]. Furthermore, the renewed commitment was prompted by the declining financial support to the extension systems and their poor performance in enhancing agricultural productivity, especially among smallholder farmers [5]. In the same period, development partners, economic advisers, and agricultural professionals proposed the privatization of extension services as a remedy to the decreased support in most SSA countries [3,6,7].

The rationale of privatizing extension services in SSA was to ease the burden of funding by the state by ensuring that the end-users pay for the cost of service delivery [8]. A financially sustainable private sector-driven extension system was expected to evolve particularly if farmers were satisfied with the quality of extension services received and would therefore be willing to pay for them [9,10]. The envisaged advantages of privatized extension services include reduced government spending, increased efficiency of service delivery, and provision of specialized programs that are flexible and responsive to site-specific needs and problems of clients [11,12].

In Liberia, the public agricultural extension service was introduced by the Ministry of Agriculture in the 1960s mainly in a top-down, supply-driven approach [13]. The system eventually collapsed during the 14-year civil war period (1989–2003) and attempts to revive it failed due to a lack of funding [14]. Recognizing the critical role of agricultural extension services in technology transfer and agricultural transformation, the government formulated the Agricultural Extension and Advisory Service (AEAS) policy in 2012 [15]. The objective was to make farmers contribute to the cost of services provided and expand the scope of the services across the country. With this policy, three types of extension services providers emerged: public (the Ministry of Agriculture), private, and non-government organizations (NGOs) [16]. While the public extension service has increasingly faced severe funding constraints and low human capital over the past few decades, the private extension service providers focus mainly on commercial farmers, with limited coverage in the rural areas [17]. Consequently, NGOs are the main providers of extension services to smallholder farmers in the areas not covered by either the state or the private sector [17].

The rapid increase in global food prices underscore the need for transformation of the agro-food industry through the provision of high quality extension services targeting smallholder farmers in developing countries [18]. Particularly in SSA countries, the renewed interest has created additional incentives for private firms to invest in extension service delivery targeting smallholder farmers [19]. According to [19,20], private extension services involve the delivery of a series of farm-based advisory services to farmers by private agents on a fee-for-service basis. The approach is expanding globally as an alternative to extension services funded by the public and contributes to reducing government spending [20]. While private extension service providers deliver services based on profit incentives, farmers' willingness-to-pay for those services depends on the benefits they expect to receive from them [21]. For instance, agricultural extension services provided by private firms can contribute to enhanced capacity of farmers to adopt and use new technologies, improve productivity to maximize profit and farmers' livelihood [22,23]. Moreover, farmers find private extension services to be more effective, efficient, and client-oriented because the agents are motivated and flexible in addressing the concerns of farmers [24].

Agriculture is the main backbone of Liberia's economy where it contributes 33 percent of the gross domestic product (GDP), employs 70 percent of the labor force, and is the sector mainly predicted by the government to pull the majority of the population out of poverty [25–27]. Rice is the main staple crop in the country and accounts for about 22% of the agricultural GDP [28]. Smallholder farmers, who account for about 71% of the rice produced [29,30], are constrained by poor access to improved inputs and farm credit with dysfunctional markets and limited knowledge for increasing production, reducing post-harvest losses, and adding value to their produce [17]. As a result, smallholder rice yields in Liberia average 1.2 metric tons per hectares (MT/ha) [31]. To overcome these and other constraints, and given the limited state funding, there is a need for the private extension services to provide a great opportunity to increase rice productivity among smallholder rice farmers in the rural areas of the country. Currently, it is neither clear if smallholder rice farmers are indeed willing to pay for private extension services nor the maximum amount they would be willing to pay for those services.

Willingness-to-pay (WTP) studies have been used to determine the perceived economic value for non-market goods or services [32]. The results are subsequently applied to determine prices of such goods and services and to evaluate their desirability by po-

tential users [33]. For example, [34] found that farmers in Northern Benin expressed WTP for extension services that would enhance their capacity to adapt to new methods. In Bangladesh, [7] found that farmers were willing to pay for extension services that would increase their economic benefits. Likewise in Tanzania, Ethiopia, and Ghana, farmers expressed their WTP for extension services with farming experience, income, age, farm size, sex, household size, and education being the major determinants of WTP [21,32,33,35]. In Liberia, however, maximum WTP for private extension services by smallholder rice farmers is not known, and the factors likely to determine it have not yet been identified. Therefore, this study assessed the smallholder rice farmers' WTP for private extension services and examined the determinants of their WTP for the service in the Gibi District of Liberia. Knowledge of the mean WTP for private agricultural extension services and the factors that determine it will enable policymakers and extension stakeholders to evaluate the viability of demand-driven extension services in Liberia. Furthermore, it will fill the existing gap in knowledge and inform possible pricing of extension services based on the attributes that smallholder rice farmers would be looking for in such a service.

## 2. Materials and Methods

### 2.1. Study Area

This study was conducted in Gibi District, Margibi County of Liberia. The district covers approximately 17,000 km$^2$ with an altitude of 30 m above sea-level, is swampy along rivers and creeks, and receives very high rainfall ranging between 4400 mm and 4500 mm per year [36]. The main vegetation consists of evergreen and deciduous trees. Agriculture is the main income-generating economic activity in Margibi County dominated by smallholder mixed food and cash crop farming. Among the food crops produced are rice, cassava, corn, and vegetables while rubber, cocoa, coffee, and oil palm are the most common cash crops produced in the district [37]. For food crops, the predominant farming practice is the traditional slash-and-burn with annual bush fallowing. Food crops are usually grown on an average farm size of 1.2 hectares and more than 75% of the produce is consumed in household [37,38]. Moreover, small-scale farmers in the study area depend mainly on NGO-provided agricultural extension services. Despite the agricultural potential of the district, it is a difficult zone to access physically for fee-for-service extension providers.

### 2.2. Theoretical Framework

In situations where goods or services are not yet available in the market, the Theory of Planned Behavior (TPB) is an important framework for analyzing potential consumers' WTP [39,40]. In a nutshell, the theory posits that behavioral intention is a function of the attitude of a consumer toward the behavior, subjective norm, and perceived behavioral control [39]. According to [39], the behavioral intention specifies the motivational factors that influence behavior and indicates an individual's readiness to perform a given behavior. Therefore, behavioral intention precedes attitude towards the behavior, which the individual evaluates as favorable or unfavorable. The TPB has been variously applied to examine the sustainable extension of food preparation techniques in rural areas of Taiwan [41] to predict WTP for municipal solid waste management in Beijing [42], and to assess the health safety behavior of farmers in Turkey [43].

From an empirical perspective, past studies have employed the double-bounded contingent valuation method (CVM) to operationalize the TPB [41,43]. The advantge of using the CVM to operationalze the TPB derives from its ability to model real-world phenomena through the creation of contigent or hypothetical scenarios that are offered to potential consumers to express their WTP [44–46]. The underlying potential motivation for a farmer's WTP for the non-marketed good or service (that is, the behavioral intention) is to maximize the utility derived from the proceeds of his investment. Considering that utility is unobservable, the researcher can only infer the potential consumer's WTP from his behavioral intention, i.e., from the verbalization of his WTP for the non-marketed good or

service [47]. Following [48], the potential consumer's initial welfare level can be expressed as an indirect utility function:

$$v\left(q^0,\ p,y\right),\tag{1}$$

where $v$ is the utility obtained from the consumption of the non-market good or service; $q_0$ is the level of welfare accruing to the potential consumer from the traditional practice, i.e., the status quo; $p$ is the price or value of forgone alternative; and $y$ is potential consumer's income. If the potential consumer has a favorable behavioral intention towards purchasing the non-market good or service once it comes to the market, then the indirect utility function becomes [49,50]:

$$v\left(q^1,\ p,y\right),\tag{2}$$

where $q^1$ denotes the proposed welfare change if the potential consumer is willing to pay for non-market commodity while the other variables are as previously defined. If the potential consumer perceives that his intended action will be beneficial to him, then the underlying utility derived from the intended action will be greater than that accruing from the status quo. Accordingly, $q_1 > q_0$, and the maximum WTP can be expressed using the compensating variation as [48]:

$$v\left(q^1,\ p,y-WTP,\ x\right) \ge v\left(q^0,\ p,y,\ x\right),\tag{3}$$

where $WTP$ is amount deducted from potential consumer's income to meet the payment for the non-market good or services and $x$ is a vector of his socio-economic characteristics. Solving for WTP in Equation (3) yields the WTP function [51]:

$$WTP = F\left(p,q^0,q^1,y,x,\ \varepsilon\right),\tag{4}$$

where $F(.)$ is the standard normal cumulative density function while the augments are as previously defined except $\varepsilon$, which is an unobservable stochastic component assumed to be normally distributed with zero mean and variance $\sigma^2$. The interpretation of the equation is that $WTP$ is the amount of money the potential consumer would be willing to pay to obtain a welfare-enhancing change, $q^1$, rather than the status quo, $q^0$.

### 2.3. Elicitation Method for WTP

The most commonly used CVM to elicit potential consumers' WTP is the double-bounded dichotomous choice (DBDC) question format [52]. The DBDC CVM protocol requires that a potential respondent is presented with two monetary values (bids), an initial and a follow-up bid for expression of his WTP. According to [53], the follow-up question, which is contingent on the response to the first bid [54], attempts to reduce bias in responses and increases the precision in the resulting WTP estimates. If the individual answers "yes" to the first bid, $B_I$, the second bid, $B_H$, is offered at a higher price, i.e., $B_I < B_H$, and if the individual answers "no" to the first bid, the second bid, $B_L$, is offered at a lower price, i.e., $B_L < B_I$ [55]. Thus, four possible outcomes (yes–yes, yes–no, no–yes and no–no) are realized for the different bid responses [53]. The likelihood of the four possible outcomes is denoted as $\pi_i^{yy}$, $\pi_i^{yn}$, $\pi_i^{ny}$, and $\pi_i^{nn}$, respectively. However, one of the weaknesses of the CVM method is that the respondents may likely state a value above their true WTP for the good or service offered to them because they do not face the obligation to actually pay the expressed WTP amount [56].

Assuming that the potential consumer is a utility maximizer, the probability of his WTP for non-market good or service is given as [55,57]:

(a)   when both answers are "yes" "yes", $B_H > B_I$, then:

$$\pi_i^{yy}(B_I, B_H) = \Pr(B_I < WTP \ge B_U) = \Pr(B_H \le maxWTP) = 1 - G(B_H, \theta);\tag{5}$$

(b)   when the first answer is "yes" followed by a "no", $B_I > B_H$, then:

$$\pi_i^{yn}(B_I, B_H) = \Pr(B_I \leq WTP \geq B_U) = G(B_H, \theta) - G(B_H, \theta); \tag{6}$$

(c)   when the first answer is "no" followed by a "yes", $B_I > B_L$, then:

$$\pi_i^{ny}(B_I, B_L) = \Pr(B_I > WTP \leq B_L) = G(B_I, \theta) - G(B_L, \theta); \tag{7}$$

(d)   when the first answer is "no" followed by a "no", $B_I > B_L$, then:

$$\pi_i^{nn}(B_I, B_L) = \Pr(B_I \geq WTP < B_L) = \Pr(B_L > maxWTP) = G(B_H, \theta), \tag{8}$$

where $\pi^{\cdot}$ is the binary indicator variable for each response (yes–yes; yes–no; no–yes; and no–no), WTP is the potential consumer's maximum WTP value, $G(B, \theta)$ is the cumulative density function of the individual's actual maximum WTP value, and $\theta$ is a vector of unknown parameters to be estimated. Given the expressions in Equations (3)–(6), the corresponding log-likelihood function for the DBDC model can be expressed as [55]:

$$\mathrm{Ln}L^d(\theta) = \sum_{n=1}^{N} \{d^{yy}\ln\pi^{yy}(B_I, B_H) + d^{yn}\ln\pi^{yn}(B_I, B_H) + d^{ny}\ln\pi^{ny}(B_I, B_L) + d^{nn}\ln\pi^{nn}(B_I, B_L)\}, \tag{9}$$

where $d^{\bullet}$ denotes the binary indicator variables that have the value of one or zero based on the individual responses, $N$ is the total sample, and $B_j$ and $\pi^{\cdot}$ are defined in Equations (3)–(6). Following [44,57,58], the double-bounded logit model was used to assess the determinants of potential consumers' WTP for non-marketed services, which in this study comprised private extension services. The model is expressed as follows:

$$WTP_i = X_i\beta + \varepsilon_i, \tag{10}$$

where $WTP_i$ is the $i$th consumer's WTP for non-market good or service, $X_i$ is the vector of explanatory variables, $\beta$ is the vector of unknown parameters to be estimated, and $\varepsilon_i$ is the error term assumed to be normally distributed. References [44,59] show that the mean WTP can be calculated once the parameters of the double-bounded logit model are estimated using:

$$E(WTP) = \bar{x}'\hat{\beta}, \tag{11}$$

where $\bar{x}'$ is the vector of the sample averages of the explanatory variables and $\hat{\beta}$ is the vector of estimated parameters.

### 2.4. Bid Schemes Use in the Double-Bounded Contingent Valuation Survey

To elicit rice farmers' maximum WTP for private extension services in Gibi District, focus group discussions (FGDs) were held in each of the three towns to generate baseline bid values as the payment vehicle to develop the contingent valuation (CV) scheme. The researcher explained the purpose of the study to the participants and then asked them to write down the maximum amount they would be willing to pay for privatized extension services. Based on the outcome of the FGDs, three bid prices (US$4.00, US$6.00, and US$12.00) were determined as payment per farm visit for Yanquilee, Peter's Town, and Wohn, respectively. Following [58], the follow-up bids were determined by presenting twice the first bid if the respondent said "yes" or half if he/she said "no". If he/she accepted the initial bid, the follow-up bids were as follows: US$8.00, US$12.00, and US$24.00, and if respondent rejected it, the follow-up bids were as follows: US$2.00, US$3.00, and US$6.00. The bid values were randomly presented to the respondents to avoid starting point bias during the survey.

### 2.5. Empirical Model

The following empirical model was fitted into the WTP data:

$$WTP_i = \beta_0 + \beta_1 AGE + \beta_2 GENDER + \beta_3 SCHL + \beta_4 EXP + \beta_5 HHSIZE + \beta_6 INC + \\ \beta_7 LAND + \beta_8 DISTEX + \beta_9 CROPDIV + \beta_{10} MBPHONE + \varepsilon_i, \tag{12}$$

where the dependent variable, $WTP_i$, was measured by the two bid values and their responses. The explanatory variables hypothesized to influence WTP for private extension services were obtained from previous studies on WTP for extension services [21,32,34,53,54] and are presented in Table 1.

**Table 1.** Explanatory variables used in the model and their hypothesized signs.

| Variable Code | Variable Name | Description | Expected Signs |
|---|---|---|---|
| AGE | Age | Age of household head in years | − |
| GENDER | Gender | Gender of household head (1 = Male; 0 = Female) | ± |
| SCHL | Years in school | Formal education of the household head in years | + |
| EXP | Farming experience | Experience in rice farming in years | + |
| HHSIZE | Household size | Number of persons depending on household head | + |
| INC | Total Annual income | Annual income of the household head in US dollars | + |
| LAND | Secured land ownership | 1 = if household owns land; 0 = otherwise | + |
| DISTEX | Distance | Distance to the extension source kilometers | − |
| CROPDIV | Crop diversification | 1 = if household diversified crop; 0 otherwise | + |
| MBPHN | Mobile Phone Ownership | 1 = if household owns mobile phone; 0 = otherwise | + |

Before fitting the model into the data, both multicollinearity and heteroscedasticity were tested. The variance inflation factor (VIF) of less than 2 indicated absence of multicollinearity among the regressors, in keeping with [60]. Likewise, the result of the Breusch-Pagan/Cook-Weisberg test showed no evidence of heteroscedasticity in the data.

### 2.6. Sampling and Data Collection

The multistage sampling technique was used to select smallholder rice farmers for the survey. In the first stage, Margibi County was purposively selected for being one of the counties in Liberia where smallholder farmers are predominantly engaged in rice production and rely largely on NGOs for extension services. In the second stage, Gibi District was selected because it is the main rice production zone in Margibi County and does not have fee-for-service extension providers. In the third stage, three townships, Peter Town, Wohn, and Yanquilee, with a high population of smallholder rice farmers in Gibi District were purposively selected. Finally, 296 smallholder rice farmers were randomly selected from a list of rice farmers in the three towns.

The study used primary data, which were collected in two phases. In the first phase, three FGDs were held with separate groups of farmers comprising youth, male and female adults, elders, and heads of farmer groups from villages in the three townships, to generate baseline bid values for use in the CV questions. In the second phase, data were collected from the 296 randomly selected heads of household in May 2019 using a pretested semi-structured questionnaire administered by trained enumerators. The questionnaire captured data on farmers' socio-economic characteristics as well as CV questions for eliciting their WTP for private extension services in rice production.

## 3. Results and Discussion

### 3.1. Farmers' Socio-Demographic Profiles

As shown in Table 2, the majority (82.4%) of the heads of household were male, reflecting the underlying patriarchal nature of the Liberian society, as indeed in much of SSA [61,62]. The mean age of the household heads was 44 years, and significantly different between households that were willing and those which were not willing to pay for private extension services. The mean age of the farmers in Gibi District attests to the fact that they were still energetic and active in their farming business, which contrasts with studies

in other SSA countries where farmers have been reported to be older [43,49]. According to [63], only farmers below the age of 50 possess the strength to carry out farming works. On average, farmers had 15 years of rice growing experience but low literacy. Low literacy compromises farmers' ability to adopt new technologies and make optimal use of their resources. The age and years of schooling agree with the national averages reported in [64].

**Table 2.** Summary statistics of farmers' socio-demographic characteristics in Gibi District.

| Variable | Willing (n = 233) | Not-Willing (n = 63) | Pooled (n= 296) | T-Value |
|---|---|---|---|---|
| *Means* | | | | |
| Age of household head (Years) | 43.4 | 46.6 | 44.1 | 2.02 ** |
| Years of schooling (Years) | 4.5 | 3.7 | 4.3 | −1.22 |
| Farming experience (Years) | 14.7 | 16.2 | 15.0 | 1.03 |
| Household size (No.) | 6.9 | 5.8 | 6.6 | −3.19 *** |
| Annual income (US$) | 1017.6 | 729.7 | 956.3 | −3.22 ** |
| Distance to extension source (Km) | 3.8 | 4.6 | 4.0 | 2.11 ** |
| *Frequencies* | | | | |
| | | Percentage | | Z-ratio |
| Gender of household head (1 = Male) | 82.0 | 84.1 | 82.4 | 0.40 |
| Land ownership (1 = Yes) | 76.4 | 77.8 | 76.7 | 0.23 |
| Crop diversification (1 = Yes) | 96.7 | 88.9 | 94.3 | −2.06 ** |
| Ownership of mobile phone (1 = Yes) | 54.1 | 49.2 | 53.0 | −0.69 |

Note: *** and ** denote 1% and 5% significance levels, respectively.

On average, households earned US$956.3 annually with significantly higher values among households that were willing to pay for private extension services. This could be attributed to engagement of willing households in other livelihood activities such as cash crop production, which also requires extension services. The result agrees with [35] who reported that higher income increases farmers' WTP for agricultural support services.

The farmers who were not willing to pay for private extension services were on average 4.6 km away from the extension service provider relative to the willing ones. Although increasing distance from extension source decreases farmers' WTP [56], a distance of 4.6 km is not too far away to prevent the unwilling households from seeking extension services. As expected in smallholder agrarian communities, the majority of the farmers in Gibi District practiced crop diversification, perhaps to hedge against the risk of crop failure to shore up household food security. More than half of the households owned a mobile phone and over three-quarters owned land. The difference in proportion was not significant between willing and unwilling households.

*3.2. Mean WTP for the Private Extension Services in Gibi District*

The mean WTP was estimated while controlling for potential confounding. The results show that 78.7% of the rice farmers were willing to pay. On average, farmers expressed a WTP of US$11.21 per farm visit for privatized extension services, which is much higher than the US$1.00 and US$1.34 per farm visit currently charged by service providers in Bangladesh and Tanzania respectively [7,36]. This finding implies that there is a high potential demand for extension services among smallholder rice farmers in Gibi District. Furthermore, it demonstrates rice farmers' eagerness to embrace the fee-for-service extension program. Our mean WTP for private extension services is lower than US$17.00 reported by [65] in Ghana and US$26.99 reported by [66] among yam growers in Nigeria. However, it is almost twice as high as the US$6.35 reported by [67] among crop farmers in Egypt. According to [11], WTP varies among geographical locations and based on information needs of the farmers and type of crops. Hence, the mean WTP established by this study falls within the range reported in other studies in SSA.

### 3.3. Determinants of Smallholder Rice Farmers' WTP for Private Extension Services in Gibi District

As shown in Table 3, the *p*-value of the Wald Chi-Square ($\chi^2$) of the double-bounded logit was 0.000, implying a good model fit. Out of ten explanatory variables, five were statistically significant. The factors that positively determined farmers' WTP for private extension services were years of formal schooling, household size, and annual income while age of the household head and distance to extension source had a significant but negative influence.

**Table 3.** Determinants of farmers' WTP for private extension services in Gibi District.

| Variables | Coefficient | Standard Error | Z-Value |
|---|---|---|---|
| Age of household head (Years) | −0.153 ** | 0.077 | 0.046 |
| Gender of household head (1 = Male) | −1.667 | 1.824 | 0.361 |
| Years of formal schooling (Years) | 0.434 *** | 0.155 | 0.005 |
| Experience (Years) | 0.027 | 0.084 | 0.752 |
| Household size (No.) | 1.080 *** | 0.294 | 0.000 |
| Annual Income ($US) | 0.003 *** | 0.001 | 0.008 |
| Land Ownership (1 = Yes) | −0.514 | 1.596 | 0.747 |
| Distance to extension provider (Km) | −0.482 * | 0.264 | 0.068 |
| Crop Diversification (1 = Yes) | 2.079 | 2.747 | 0.449 |
| Ownership of mobile phone | 1.847 | 1.353 | 0.172 |
| Constant | 11.206 | 4.755 | 0.018 |
| Wald Chi$^2$ (10) = 39.77 | | | |
| Prob. > Chi$^2$ = 0.000 *** | | | |
| Log-likelihood = −332.39447 | | | |
| Number of Observations 296 | | | |

Note: ***, **, * denote 1%, 5%, and 10% significance levels, respectively.

As expected from theory (see Table 1), the age of the household heads was negatively related to WTP for private extension services. This is logical because older farmers have more farming experience acquired through years of trial and error and would therefore not be willing to invest in extension services that add little (at best) to their existing stock of knowledge. The negative relationship implies that an additional year in farmers' age would lead to a 0.15 unit decrease in farmers' WTP for private extension services. This finding is consistent with that of [68] who reported that a negative relationship between farmers' age and their WTP for insect-based feed for fish and pigs in Kenya.

Years of formal schooling were positively and significantly associated with WTP for private extension services as expected *a priori*. Accordingly, an additional year in formal schooling would increase a farmer's WTP by 0.43 unit. A plausible explanation is that educated farmers are more likely to understand, interpret, and apply new information they receive from private extension service providers. The result tallies with that of [36], who reported a positive influence of education on farmers' WTP for sustainable agricultural land use in the GAP-Harran Plain of Turkey.

As expected *a priori*, household size was positive and significantly related to rice farmers' WTP for private extension services. Hence, a unit increase in household size will increase farmers' WTP for extension services by 1.1 units. The result suggests that large households were more willing to pay for private extension services than smaller ones. Large households probably require more food and, hence, would invest in private extension services to increase production to meet both household consumption requirements and the income generation goal. This result contradicts that of [21] that reported a negative relationship between household size and farmers' WTP for privatized agricultural extension services in Tigray, Ethiopia. However, it tallies with [69], who found a positive relationship between household size and WTP for irrigation water among rice farmers in Uganda.

Annual income was positive and significantly related to farmers' WTP for private extension services in Gibi District, as expected from theory. Accordingly, a unit increase in income would lead to a 0.003 unit increase in WTP, confirming the positive income effect for a normal good/service in consumer demand theory (e.g., see [70]). With an average income of US$287.90 above the unwilling households, the households that were willing to pay for private extension services also had the ability to do so. In other words, the mean WTP and the mean annual income suggest that the smallholder rice farmers in Gibi District had both the willingness as well as the ability to pay for private extension services. This finding is consistent with [71], who reported a positive association between income and farmers' WTP for extension services in the GAP-Harran Plain in Turkey.

The distance to the nearest extension source was negatively related to farmers' WTP for private extension services as expected *a priori*. Accordingly, a one unit increase in distance to the nearest extension service provider would decrease farmers' WTP for private extension services by 0.48 unit. This could be attributed to high transaction costs associated with both the physical travel and time cost to and from the extension service provider. These transaction costs escalate when one considers the deplorable state of roads in the rural areas of Liberia. Therefore, farmers find it more expensive and time-consuming to travel to procure extension services. On the other hand, it is also costly for farmers in distant areas to cover the transport cost for service providers to travel to their farms compared to those closer to the sources. The result agrees with [35], who reported a negative relationship between distance to the extension service provider and farmers' WTP for agricultural extension services in the Mpwapwa and Mvomero districts of Tanzania.

## 4. Conclusions and Recommendations

The study was carried out to estimate smallholder rice farmers' mean WTP for private agricultural extension services as well as to assess the determinants of that WTP in Gibi District of Liberia. A multistage sampling technique was used to select 296 smallholder rice farmers, and DBDC CVM was used to elicit the farmers' maximum WTP for private extension services. Descriptive statistics and the double-bounded logit were used to analyze the data. The findings revealed significant differences in the socio-economic characteristics of the farmers who were willing to pay compared to those who were not. For example, those who were willing to pay were younger, had spent more years in formal schooling, and earned higher annual income than their counterparts. In contrast, the households that were not willing to pay had fewer family members and lived furthest from the extension providers, which escalated their extension service seeking costs. Most (78.7%) farmers in the sample were receptive to the idea of paying for extension services, which bodes well with the Government of Liberia's desire to privatize agricultural extension owing to high fiscal constraints. The study also found a relatively high potential demand for private extension services as evidenced by a mean WTP of US$11.21 per farm visit. While this mean is almost twice the average wage of local skilled worker of US$6.00 per day, it falls within the range of mean WTP values reported in studies in other SSA countries. Nevertheless, the high mean WTP found in this study suggests some potential strategic bias on the part of the respondents given that they are not faced with the actual obligation to pay. Therefore, the mean WTP reported in this study should be used with caution.

The results of the double-bounded logit confirmed the underlying demand theory. Of note is the positive income effect that suggests that agricultural extension services in Gibi District are normal "goods". Both the relatively high mean WTP and mean annual income suggest that smallholder rice farmers in the district have the wherewithal to express their effective demand by matching their WTP with their ability to pay for private extension services. High travel and time costs diminish farmers' WTP for private extension services, as expected, and could escalate if the deplorable state of roads in rural Liberia is not quickly addressed.

The study, therefore, recommends that given the many development priorities amidst limited funding for agriculture, the Government of Liberia should encourage the private

sector to invest in extension services to take advantage of the high farmers' WTP. Service providers should target younger farmers since they are more willing to pay for extension services compared to the older ones. This can be done by including innovative learning approaches, such as information communication technology (ICT) and e-extension services that are currently being piloted in many SSA countries. Furthermore, given the low literacy level among the sample farmers, care should be taken to avoid marginalizing or excluding them by designing and implementing adult-related education programs that suit their needs. Additionally, considering the negative influence of the distance to extension providers, the government should look for funds to rehabilitate rural feeder roads to connect farmers to markets and service providers at reduced transaction costs. Finally, the government of Liberia and its development partners should design and implement programs that will increase farmers' income to enhance their capacity to pay for privatized extension services.

**Author Contributions:** Conceptualization, T.V.S. and C.R.; methodology, T.V.S. and P.I.; software, T.V.S.; validation, T.V.S., C.R. and P.I.; formal analysis, T.V.S.; investigation, T.V.S.; resources, T.V.S.; data curation, T.V.S., C.R. and P.I.; writing—original draft preparation, T.V.S.; writing—review and editing, C.R. and P.I.; visualization, T.V.S.; and supervision, C.R. and P.I. All authors have read and agreed to the published version of the manuscript.

**Funding:** This study was supported by the African Economic Research Consortium (AERC).

**Institutional Review Board Statement:** Not applicable.

**Informed Consent Statement:** Informed consent was obtained from all subjects involved in the study.

**Data Availability Statement:** The data presented in this study are available on request from the corresponding author.

**Acknowledgments:** The authors are grateful to the Almighty God for the strength and guidance to write this paper. We extend our appreciation to Faculty members and students at the Department of Agricultural Economics, University of Nairobi, for their contributions to this paper. Finally, a sincere appreciation to the African Economic Research Consortium (AERC) for supporting the study reported in this paper.

**Conflicts of Interest:** The authors declare no conflict of interest. The funders had no role in the design of the study; in the collection, analysis, or interpretation of data; in the writing of the manuscript; or in the decision to publish the results.

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
