# Peer review of "Determinants of Smallholder Rice Farmers’ Willingness-to-Pay for Private Extension Services in Liberia: The Case of Gibi District"

_sustainability, doi:10.3390/su151914300_

Round 1
Reviewer 1 Report
The comments and suggestions are attached to it.

the manuscript needs a minor revision.
Author Response
Response to Reviewer 1 Comments
Point 1: you are at the same department, why you put number 2 , just star for the correspondent author is fine.
Response 1: The reviewer’s point was well noted and considered. Thus, the numbering was removed and a star used to indicate the corresponding author.
Point 2: If you are belonging to the Nairobi university, then it’s better to use email from university domain.
Response 2: The author, a graduate of the University of Nairobi, does not have access to his student account. Therefore, he is using his personal email address.
Point 3: Please follow the format submission which explained in the sustainability link, for all your paper.
Response 3: The reviewer’s request to follow the format for sumission for intext citation valid and the needful was done across the entire paper.
Point 4: Its really interesting to combine this method with CVM, It will be notable that if you give more citeable reference regarding that.
Response 4: additional references were included in the section on CVM.
Point 5: It will be great if you give more detail about FGD, regarding your procedure(howmany, for each fgd how many were participate?).
Response 5: More details are provided on the FDG. The prodedure was explained interms of the number of particpants in each session held including the composition of the participants – male, female, youths, etc.
Point 6: why pretest?.
Response 6: The questionnaire was preseted because the authors wanted to examine if the potential respondents understood the wordings their meaning and that there is clearity in the strructure of the questions. Additionally, the sentence containing pretest was revised thus including the purpose for pre-testing the questionnaire.

Reviewer 2 Report
The article generally explores the feasibility of smallholder rice farmers using private extension services in Liberia. The discussion and the overall contents are fair enough but the Turnitin report shows 81% similarity from other sources. Require major rephrasing or correction for related content and make sure its similarity is lower than 30%.
Author Response
Response to Reviewer 2 Comments
Point 1: The article generally explores the feasibility of smallholder ricefarmers using private extension services in Liberia. The discussion and the overall contents are fair enough but the Turnitin reportshows 81% similarity from other sources. Require major rephrasingor correction for related content and make sure its similarity islower than 30%..
Response 1: The manuscript is an output of my MSc thesis which was written in a paper format [chapter 5 (with the intent to publish each objective/chapter)]. The thesis was also published in the University of Nairobi repository, thus contributing to the high similarity score. See the link to the published thesis:
http://erepository.uonbi.ac.ke/handle/11295/160305.
However, prior to the publication of the thesis, the overall similarity score for the thesis was 14%. I also shared the plagiarism score of the MSc thesis (signed electronically by Dr. Ritho, co-author of the paper and 1st supervisor of the thesis.) with the Assistant Editor, Justyna Hałabuza.

Reviewer 3 Report
The topic of the article is very current and interesting. The article examines the topic only on a regional scope, therefore it is necessary to describe the current state of agricultural production into more details, including the social structure of farmers in the selected region. The article lacks a precise definition of "private extension services" and what benefits they provide to farmers. Authors do not respect the citation format required by the editors which needs to be corrected.
Particular comments:
Introduction
· Lines 96 – 98: delete the phrase „The rest of the paper is organized as follows:……………..
· Define a scientific hypotheses and partial objectives/goals.
Materials and Methods
· Subchapter 2.1 Theoretical Framework – this part should be moved and included to Introduction
· Subchapter 2.5. Study Area – this part should be moved and used as the subchapter „2.1“
· 2.6. Sampling and data collection – it is notcompletely clear how many respondents were used for the analyses
Results and Discussion
· It is not clear (sufficiently described) what benefits the farmers will gain
· There is no discussion concerning the issue whether extension services can lead to dependence on the supply of specific products (pesticides, fertilizers)
· There is no discussion concerning the relationship between provided agricultural extension Services and the environmental safety.
Conclusions and Recommendations
· Formulate the conclusions according to the partial goals/objectives and the newly set hypothesis
In conclusion section, how do you answer your goal/objective "…..the attributes the farmers are looking for in such a service."?
Author Response
Response to Reviewer 3 Comments
Point 1: The topic of the article is very current and interesting. The article examines the topic only on a regional scope, therefore it is necessary to describe the current state of agricultural production into more details, including the social structure of farmers in the selected region. The article lacks a precise definition of "private extension services" and what benefits they provide to farmers. Authors do not respect the citation format required by the editors which needs to be corrected.
Response 1: We thank the reviewer for the well detailed feedback provided on the paper. We have provided a description of the current state of production and the social structure of farmers in the region under section 2.1 (Study Area). Additionally, a paragraph on the definition of private extension services and what benefits they provide to farmers were added in the intrudction. On the citation format, as was also highlighted by one of the reviewers, all intext citations have been revised in line with the journal’s requirement.
Point 2: Lines 96 – 98: delete the phrase „The rest of the paper is organized as follows:……………..
Response 2: The phrase was deleted as recoemmended.
Point 3: Define a scientific hypotheses and partial objectives/goals.
Response 3: The authors clearly stated (revised) the objective in the manuscrip as was highlighted by the review.
Furthermore, following common practices in most research articles, since the objective is already stated, therefore, the hypothesis implies as, “Farmers’ socioeconomic characteristics are key determinants of WTP for private extensin services in the Gibi District”.
Point 4: Subchapter 2.1 Theoretical Framework – this part should be moved and included to Introduction
Response 4: In the view of the authors and in accordance with previous valuation studies – for instance, OU and Barry (2018) and Gbenou-Sisinto (2018) published by Sustainability – maintained the Theoritical Framework that underpins the study under the material and method session of their paper. This is because the theory forms part of the methodology thus connecting well with the model. Furthermore, keeping it under the material and method session allows readers to have a clear understanding of the approach used. Moving it to the introduction session would sound more like a literature review that supports the background or basis for the study. Therefore, we kindly appeal to the reviewer for reconsideration on this suggestion in this point.
Point 5: Subchapter 2.5. Study Area – this part should be moved and used as the subchapter „2.1“
Response 5: The subchapter on study area (2.5) was moved to 2.1 and the necessary adjustment was made on the numbering of the preceeding subchapters.
Point 6: 2.6. Sampling and data collection – it is not completely clear how many respondents were used for the analysis.
Response 5: We have included the total sample size of 296 in subchapter 2.6 of the study.
Point 7: It is not clear (sufficiently described) what benefits thefarmers will gain
Response 7: Considering the valuation nature of the study, the service providers and policy makers are the direct beneficiaries of the outcome of the study, not the farmers. However, as service providers are atrracted to invest in private extension services because of it viability / the farmers expressed willlingness to pay for it, it is assumed that they (service providers) will make their services more accessible to the farmers. Hence, the benefit to the farmers shall be the availability and accessibility of extension serivices to all farmers since they are willing to pay for it. Meanwhile, the benefits of private extenison is higlighted in the intrudction in line with point 1.
Point 8: There is no discussion concerning the issue whether extension services can lead to dependence on the supply of specific products (pesticides, fertilizers)
Response 8: The concern is welcoming. However, we could not explore whether extension servicesd can lead to dependence on the supply of specific products. This is because, the focus/scope of the study was to valuate private extension services. Additionally, we do not have the data that links what the reviewer has suggested. Therefore, we would not delve into that.
Point 9: There is no discussion concerning the relationship between provided agricultural extension Services and the environmental safety.
Response 9: The authors appreciate the reviewer for this excellent observation. However, the paper could not explore the relationship between provided agriculture extension services and the environmental safety. This is soley because the goal of the paper was to assess whether farmers are willing to pay for private extension service which is not available to smallholder farmers, expecially those in rural areas. Therefore, it was not fessible to provide discussion on a relationship between the provide services and the environmental safety. It is the reseachers anticipation that if the services are rollout by private service providers, further research could explore the different services that will be demanded by the clients.
Point 10: Conclusions and Recommendations
Formulate the conclusions according to the partial goals/objectives and the newly set hypothesis
Response 10: Once again we appreciated the reviewer for the observation on the partial objective. This was revised in the introduction session. Hence, it now aligns with the construct of the conclusion. Additionally, we could not state the hypothesis (Farmers’ socioeconomic characteristics are key determinants of WTP for private extension services in the Gibi District) that was tested because it is implied as it is done in other studies.
Point 11: In conclusion section, how do you answer your goal/objective"…..the attributes the farmers are looking for in such a service."?
Response 11: In context of the study, the results have direct implication on service providers and policymakers instead of the farmers (indirect implication). The results seek to inform service providers about the willingness to farmers to pay for private extension services and how viable it is to invest in private extension services. Additionally, provides baseline information for policymakers to make inform decisions that are based on evidence to enhance the enabling environment for service providers to deliver adequate, effective and efficient services to the farmers. Moreover, the attributes or socioeconomic characteristics identified as the key determinants of willingness to pay are what service providers should look for to tailor their services for the farmers in the study area. Furthermore, farmers will look out for the expected benefits of the services based on what they need.

Reviewer 4 Report
I thank the authors for giving me the opportunity to read their interesting work.
I suggest that we
-) supplement the bibliography with other international paper
-) highlight in the results the sample size; 296 respondents mentioned only in the abstract
-) highlight the limitations of the work and possible research developments
Author Response
Response to Reviewer 4 Comments
Point 1: Supplement the bibliography with other international paper.
Response 1: Additional references are included.
Point 2: Highlight in the results the sample size; 296 respondentsmentioned only in the abstract r.
Response 2: We thank the reviewers for this observation. The sample size of 296 was included in the methodology under sample size session and in the opening sentence of the results and discussion session.
Point 3: Highlight the limitations of the work and possible research developments.
Response 3: Though we found the comment very insigtful for the study, we kindly ask the reviewer to reconsider it. This is because, the author purposely want to leave this with the reader to identify the limitation(s) of the study and the posible research development. One key limitation of the study was funding to facilitate the field work. Furthermore, researcher would have love to cover more districts or counties of Liberia as well as varity of crop farmers to assess their willingness to pay for private extension services. However, we were confined to only rice farmers and one districy. Althought our findings close the gap in knowledge on rice farmers’ WTP for private exgtension services in Liberia, woul appreciate if future research look at the technical efficiency of private extension services in Liberia.

Round 2
Reviewer 2 Report
Overall, the current version of the article is now better. Most of the corrected items are able to enhance the understanding of the clarity of the discussion. However, the reviewer would like to suggest to the author add several recent citations (2022 or 2023) to prove that the issues are still popular and attracted many researchers to perform such a study.
Author Response
Point 1: Overall, the current version of the article is now better. Most of the corrected items are able to enhance the understanding of the clarity of the discussion. However, the reviewer would like to suggest to the author add several recent citations (2022 or 2023) to prove that the issues are still popular and attracted many researchers to perform such a study.
Response 1: We thank the reviewer for acknowledging the improvement in the quality of our work as a result of the comments provided by him/her. We further thank the reviewer for observation and suggestion for current citation and wish to express that we have provided citations of recent studies up to 2023. We also made significant improvements in the clarity of the results, and the study's conclusion.
Reviewer 3 Report
It's good to see that you appreciate all the comments. Comments have been carefully addressed.
Line 95: edit, „…..tential users [30].“ - you provide a superscript
Author Response
Point 1: Line 95: edit, „…..tential users [30].“ - you provide a superscrip.
Response 1: We thank the reviewer for the observation. The superscript was removed.